# Neutrophil-to-lymphocyte ratio as a prognostic indicator in COVID-19: Evidence from a northern tanzanian cohort

Norman Jonas Kyala[1,2]*, Innocent Mboya[3,5], Elichilia Shao[1,2], Francis Sakita[1,4], Kajiru Gadiel Kilonzo[1,2], Laura Shirima[3], Abid Sadiq[1,2], Elifuraha Mkwizu[1,2], Nyasatu Chamba[1,2], Annette Marandu[1,2], Sophia Muhali[1,2], Faryal Raza[1,2], Eliasa Ndale[1,2], Damas Bayo[1,2], Daniel Mujuni[1,2], Furaha Lyamuya[1,2]

1 Faculty of Medicine, Kilimanjaro Christian Medical University College, Moshi, Tanzania, 2 Department of Internal Medicine, Kilimanjaro Christian Medical Centre, Moshi, Tanzania, 3 Department of Epidemiology and Biostatistics, Institute of Public Health, Kilimanjaro Christian Medical University College, Moshi, Tanzania, 4 Department of Emergency Medicine, Kilimanjaro Christian Medical Centre, Moshi, Tanzania, 5 Department of Translational Medicine, Lund University, Malmö, Sweden

☯ These authors contributed equally to this work.

* normanjonasmd@gmail.com

**Data Availability Statement:** All relevant data are within the manuscript and its Supporting Information files.

## Abstract

### Background

COVID-19 caused a profound global impact, resulting in significant cases and deaths. The progression of COVID-19 clinical manifestations is influenced by a dysregulated inflammatory response. Early identification of the subclinical progression is crucial for timely intervention and improved patient outcomes. While there are various biomarkers to predict disease severity and outcomes, their accessibility and affordability pose challenges in resource-limited settings. We explored the potentiality of the neutrophil-to-lymphocyte ratio (NLR) as a cost-effective inflammatory marker to predict disease severity, clinical deterioration, and mortality in affected patients.

### Methodology

A hospital-based retrospective cohort study was conducted at KCMC Hospital among COVID-19 patients followed from admission to discharge between 1st March 2020 and 31st March 2022. NLR was calculated as the absolute neutrophil count in µL divided by the absolute lymphocyte count in µL. The NLR cut-off value was determined using Receiver Operating Characteristic (ROC) analysis and assessed its predictive ability at admission for in-hospital mortality. The Chi-square test compared the proportion of NLR by patient characteristics. The association of NLR with disease severity and mortality was analyzed using the modified Poisson and Cox regression models, respectively.

### Results

The study included 504 patients, with a median age of 64 years, 57.1% were males, and 68.3% had severe COVID-19. The in-hospital COVID-19 mortality rate was 37.7%. An NLR

**Funding:** The author(s) received no specific funding for this work.

**Competing interests:** The authors have declared that no competing interests exist.

cutoff value of 6.1 or higher had a sensitivity of 92.1% (95% CI 89.2%–94.0%) and a specificity of 92.0% (95% CI 89.7%–94.4%). Additionally, 39.5% of patients with an NLR value of 6.1 or higher had increased risk of severe disease, subsequent clinical deterioration, and mortality.

## Conclusion and recommendation

An NLR value of 6.1 or higher at the time of hospital admission associated with severe disease, clinical deterioration, and mortality in patients with COVID-19. Integration of NLR as a prognostic parameter in COVID-19 prognosis scales could improve risk assessment and guide appropriate management strategies for COVID-19 patients, as well as for potential future viral-related pneumonias. Further prospective studies are necessary to validate these findings and evaluate the clinical utility of NLR in larger cohorts of patients.

## Introduction

Coronavirus disease 2019 (COVID-19), caused by the novel severe acute respiratory syndrome coronavirus 2 (SARS-CoV-2), emerged in Wuhan, China, in late 2019 and became a global pandemic in March 2020 [1–4]. Tanzania reported its first case on March 16, 2020, with 37,091 confirmed cases and 808 deaths as of March 4, 2023 [5].

The disease varies widely in clinical presentation as severe cases are at an increased risk of mortality, hence detecting clinical deterioration early is crucial [6, 7]. Evidence suggests that the Neutrophil-to-Lymphocyte Ratio (NLR) may serve as a marker for assessing disease severity, and predicting clinical deterioration and mortality in COVID-19 patients diseases [8–18]. NLR reflects the balance between neutrophils and lymphocytes in the blood, offering insights into the immune response and inflammation. It combines changes in neutrophil and lymphocyte levels, enhancing its sensitivity as an immune response marker [19–21]. Clinically, NLR can stratify patients, aiding in the identification of critically ill individuals [19–21].

From an immunological perspective, NLR mirrors the interplay between innate and adaptive immune responses during illness and pathological stress [21]. Notably, changes in the NLR often occur before the occurrence of the disease's clinical manifestations, enabling early detection and informing clinicians about the ongoing subclinical pathological processes (Zahorec, 2001). In the context of SARS-CoV-2 infection, the mechanisms underlying the response of neutrophils and lymphocytes have been postulated. Neutrophils play a crucial role in activating the immune system and releasing reactive oxygen species (ROS). These ROS can cause DNA damage in infected cells, leading to the release of the virus, which can then be targeted by antibodies. Neutrophils can also stimulate the production of various cytokines like Interleukin-8 (IL-8), Tumor necrosis factor-alpha (TNF-alpha), and Interleukin-1 beta (IL-1β), as well as effector molecules including ROS, defensins, proteases, and nitric oxide (NO). Conversely, systemic inflammation, particularly elevated Interleukin 6 levels in COVID-19, paradoxically leads to a decrease in lymphocyte count, thereby impairing cellular immunity. Both these factors contribute to an elevated NLR [8, 22, 23].

Given NLR's success in predicting disease severity in various conditions, its utility among COVID-19 patients, particularly in resource-limited settings like Tanzania, deserves exploration [8–18]. The study's importance lies in NLR's simplicity and cost-effectiveness compared to other inflammatory markers [19–21, 24]. It can be calculated from routine blood tests,

aiding in the early identification of high-risk patients when used alongside clinical methods. If backed with evidence, NLR can inform clinical guidelines and algorithms, enhancing COVID-19 management and resource allocation. This research provides relevant evidence and lays the foundation for future studies exploring NLR's broader applicability in clinical settings and monitoring COVID-19 patients [19–21, 24]. However, it is important to note that NLR has been shown to fluctuate with the administration of clinical or supportive therapies, which can significantly impact patient management strategies. For instance, the administration of systemic steroids, commonly used in the management of severe COVID-19, has been associated with alterations in leukocyte counts, subsequently influencing the NLR [25].The study evaluated NLR as a potential marker for assessing disease severity and predicting clinical deterioration and mortality among COVID-19 patients at Kilimanjaro Christian Medical Centre (KCMC) Hospital in Northern Tanzania.

## Methods

### Data source and population

This research employed a retrospective cohort study design, from March 2020 to March 2022 at KCMC zonal-referral Hospital with a catchment area of 15 million people located in Moshi Municipality, Kilimanjaro region, northern Tanzania. The hospital has a bed capacity of over 640 beds. The study population consisted of 504 adult patients aged 18 years or older hospitalized at KCMC with a confirmed COVID-19 diagnosis based on positive RT-PCR or antigen test results for SARS-CoV-2 conducted by the National Laboratory of Public Health of Tanzania. We excluded pregnant women, individuals missing baseline neutrophil and/or lymphocyte data, confirmed hematological malignancies, and receipt of chemotherapy, immunomodulating drugs, or long-term glucocorticoids, as these conditions could potentially affect neutrophil, lymphocyte, and platelet counts, leading to inaccurate NLR measurements.

Data was collected from 1st October 2022 to 31st December 2022. Data collection involved reviewing individual patient clinical files using a data extraction sheet encompassing clinical and laboratory data, including presenting symptoms, respiratory rate on admission, oxygen saturation, days from symptom onset to hospitalization, length of hospital stay, treatment outcomes, age, gender, disease severity, and documented comorbidities. Laboratory data included hemoglobin level, leukocyte count, thrombocyte count, lymphocyte count, serum creatinine, urea, aspartate transaminase (AST), and alanine aminotransferase (ALT).

### Variables

We examined several key variables to investigate the association between NLR with patient's COVID-19 outcomes. The primary outcomes were COVID-19 severity, clinical deterioration during hospital admission, and COVID-19-related mortality during hospitalization. COVID-19 severity was categorized into four levels: mild, moderate, severe, and critical, following the interim guideline of the WHO for COVID-19 (WHO, 2021). Mild cases were characterized by symptomatic patients meeting the case definition for COVID-19 without evidence of viral pneumonia or hypoxia. Moderate cases included patients with clinical signs of pneumonia but no signs of severe pneumonia, with peripheral capillary oxygen saturation (SpO2) $\geq$ 90% on room air. Severe cases were defined by clinical signs of pneumonia, a respiratory rate > 30 breaths/min, severe respiratory distress, or SpO2 < 90% on room air (WHO, 2021). Critical cases were identified by criteria for Acute Respiratory Distress Syndrome (ARDS), sepsis, septic shock, or other conditions necessitating life-sustaining therapies such as mechanical ventilation (invasive or non-invasive) or vasopressor therapy (WHO, 2021).

The primary independent variables examined were the absolute neutrophil count, absolute lymphocyte count recorded on admission, and the calculated NLR. NLR was calculated by dividing the absolute neutrophil count in in µL by the absolute lymphocyte count in µL obtained from the full blood picture of patients upon admission. NLR was analyzed both as a continuous and categorical variable, with categories of normal NLR and elevated NLR. The cut-off value for the categorized NLR was determined through the Receiver Operating Characteristic (ROC) analysi.

## Data analysis

The extracted data were transferred from the Excel spreadsheet to the STATA version 16 for processing and analysis [26]. The Chi-squared test determined the association between NLR and participant characteristics. The Kruskal-Wallis test was performed to compare the medians of NLR across groups of patients based on their disease severity and clinical deterioration. For the assessment of clinical deterioration, the analysis focused on patients admitted with mild and moderate disease, and then follow-up for clinical outcomes. The outcomes included; those who did not deteriorate, those who deteriorated but survived, and those who deteriorated and died.

In order to evaluate the diagnostic accuracy of the Neutrophil-to-Lymphocyte Ratio (NLR) in predicting mortality among COVID-19 patients, a Receiver Operating Characteristic (ROC) analysis was conducted. The ROC analysis method has been widely utilized in various studies aimed at predicting mortality among COVID-19 patients. This analysis involved plotting the ROC curve and calculating the Area Under the Curve (AUC) to evaluate the overall performance of NLR as a prognostic marker. Sensitivity and specificity values were determined at various NLR thresholds, allowing us to identify the optimal threshold that maximizes diagnostic accuracy. This optimal threshold provides a reliable basis for predicting patient mortality during admission based on their NLR values.The association of elevated NLR above the cut-off value with disease severity was examined using the modified Poisson regression model. The association of NLR above the cut-off value with mortality was investigated using Cox regression models. All regression models were adjusted for all statistical tests were two-sided at a 5% threshold level.

## Ethical consideration

Ethical approval was obtained from the Kilimanjaro Christian Medical University College Research and Ethics Review Committee with approval no PG/91/2022. Permission to extract hospital data was obtained from the Director of Hospital Services through the Head of the Department of Internal Medicine at KCMC. Unique patient identification instead of hospital medical record numbers were used to maintain confidentiality. Informed consent was waived by the Kilimanjaro Christian Medical University College Research and Ethics Review Committee for this retrospective cohort study. The waiver was granted based on the use of fully anonymised data, minimizing any risk to participants. The study adhered to ethical guidelines to ensure confidentiality and privacy of participant information.

## Results

574 confirmed COVID-19 patients were admitted to KCMC Hospital between March 2020 and March 2022, among those 504 patients were enrolled in the study after meeting the inclusion criteria. The median age of the cohort was 64 years with an interquartile range (IQR) of 53–75 years. Among these patients, 60% were at least 61 years old, and 57.1% were males, as shown in Table 1.

**Table 1. Distribution of NLR across background characteristics of COVID-19 patients admitted at KCMC Hospital between March 2020 to March 2022 (N = 504).**

| Characteristic | Total | NLR | | *P- value |
|---|---|---|---|---|
| | | NLR< 6.1 n (%) | NLR ≥ 6.1 n (%) | |
| Age (years) | | | | |
| Median (IQR) | 64 (53–75) | xx | | |
| 18–30 | 19 (3.6) | 11 (61.1) | 8 (38.9) | 0.001 |
| 31–60 | 183 (36.4) | 131 (71.6) | 52 (28.4) | |
| 61 and above | 302 (60) | 163 (54) | 139 (46) | |
| Sex | | | | |
| Male | 288 (57.1) | 172 (59.7) | 116 (40.3) | 0.674 |
| Female | 216 (42.9) | 133 (61.6) | 83 (38.4) | |
| COVID-19 Waves | | | | |
| Wave 1 | 111 (22.0) | 80 (72.1) | 31 (27.9) | 0.036 |
| Wave 2 | 190 (37.7 | 105 (55.3) | 85 (44.7) | |
| Wave 3 | 119 (23.6) | 70 (58.8) | 49 (41.2) | |
| Wave 4 | 84 (16.7) | 50 (59.5) | 34 (40.5) | |
| Comorbidities (Yes) | | | | |
| Diabetes mellitus | 160 (31.8) | 93 (58.1) | 67 (41.9) | 0.454 |
| Hypertension | 260 (51.6) | 154 (60.5) | 106 (40.5) | 0.542 |
| HIV/AIDS | 13 (2.6) | 9 (69.2) | 4 (30.8) | 0.232 |
| Chronic kidney disease | 34 (6.8) | 19 (55.9) | 15 (39.1) | 0.576 |
| Other comorbidities | 14 (2.8) | 9 (64.3) | 5 (36.7) | 0.563 |

Abbreviations: HIV/AIDS, Human Immunodeficiency Virus / Acquired Immunodeficiency Syndrome. NLR, Neutrophil to Lymphocyte Ratio.

*P-value from the Chi-squared distribution.

The median NLR values increased as the COVID-19 severity increased. Patients in the critical group had the highest median NLR of 13.6 (IQR 10.3–25.9), whereas patients with mild disease had a median NLR of 2.1 (IQR 1.7–3.4), with a p-value < 0.001 (S1 Fig).

Regarding clinical deterioration during hospital stay; in patients admitted with mild and moderate COVID-19 higher NLR during admission was significantly associated with clinical deterioration and progression to poor outcomes. Specifically, for patients with mild and moderate COVID-19 during admission who did not deteriorate during their hospital stay, their median NLR was 1.9 (IQR 1.4–2.6). For those who deteriorated to severe or critical disease but were discharged alive, the median NLR was 5.6 (IQR 4.4–6.2). For patients who were admitted and deteriorated to severe/critical disease and subsequently passed away, the median NLR was 13.5 (IQR 6.4–24.5) (as shown in S2 Fig).

The NLR cut-off in predicting mortality due to COVID-19 determined by ROC analysis was 6.10 with a sensitivity of 92.1% (95% CI 89.2%–94.0%) and a specificity of 92.0% (95% CI 89.7%–94.4%). The AUC was 0.9207 (95% CI 0.896–0.945). The high sensitivity and specificity values suggest that the NLR cut-off of 6.1 can be considered as a threshold for predicting the mortality of COVID-19 patients at admission (S3 Fig).

The distribution of NLR across background characteristics of COVID-19 patients admitted at KCMC Hospital showed that out of 504 patients, 199 patients (39.5%) had a higher NLR (6.1 or above). Patients aged 61 years and above had the highest proportion of NLR of 6.1 or above, with 139 patients (69.8%) (p-value = 0.001) compared to other age groups. Although males showed a slightly higher proportion of NLR of 6.1 or above 116 (40.3%) compared to females 83 (38.4%), the difference was not statistically significant. The second wave of

**Table 2. Distribution of NLR across clinical outcomes of patients admitted with COVID-19 at KCMC hospital between March 2020 to March 2022 (N = 504).**

| Clinical outcome | Total | NLR | | *P-value |
|---|---|---|---|---|
| | | NLR < 6.1 n = 305 | NLR ≥ 6.1 n = 199 | |
| COVID-19 Severity | | | | |
| Mild | 38 (7.5) | 35 (92.1) | 3 (7.9) | <0.001 |
| Moderate | 102 (20.2) | 82 (80.4) | 20 (19.6) | |
| Severe | 344 (68.3) | 185 (53.8) | 159 (46.2) | |
| Critical | 20 (4.0) | 3 (15) | 17 (85) | |
| Clinical deterioration (n = xxx) | | | | |
| No | 90 (64.3) | 89 (98.9) | 1 (1.1) | <0.001 |
| Yes, and discharged alive | 31 (22.1) | 23 (74.2) | 8 (25.8) | |
| Yes, and dead | 19 (13.6) | 5 (26.3) | 14 (73.7) | |
| Mortality from COVID-19 | | | | |
| Dead | 190 (37.7) | 16 (8.4) | 174 (91.6) | |
| Alive | 314 (62.3) | 289 (92.0) | 25 (8.0) | <0.001 |

Abbreviations: NLR, Neutrophil to Lymphocyte Ratio

*P-value from the Chi-squared distribution.

COVID-19 had the highest proportion of patients with NLR of 6.1 or above 85 (44.7%) compared to other waves of COVID-19 (p-value = 0.04) (Table 1).

In terms of COVID-19 severity, a higher proportion of patients with mild COVID-19 had NLR below 6.1 (35 or 92.1%). It is interesting to note that among patients with severe and critical COVID-19, 159 (46.2%) and 17 (79.9%) had NLR of 6.1 and above (Table 2).

Of all 504 patients, 140 (27.8) were admitted with mild and moderate disease. Among those who deteriorated and survived 8 (25.8%) had NLR of 6.1 and above compared to 14 (73.7%) in patients who deteriorated and died (p<0.001), suggesting higher NLR to be associated with poor progression of COVID-19 (Table 2).

Among the patients who died during their hospital stay, a majority of 175 (91.6%) had an NLR of 6.1 or higher, p-value<0.001) (Table 3).

Individuals with a baseline NLR of 6.1 or above had a 1.43 times higher risk of developing severe and critical disease than their counterparts (RR 1.43; 95% CI 1.29–1.58, p-value = <0.001). In addition, in the multivariable analysis, only the second and fourth waves and hypertension exhibited a significant positive association with severe and critical disease (Table 3).

Regarding COVID-19 mortality, the adjusted analysis showed that NLR had a positive association with mortality. For every unit increase in NLR, the mortality risk increased by 1.03 (AHR 1.03; 95% CI 1.02–1.04, p-value < 0.001). Other factors associated with higher mortality included age >60 years, longer duration since the first symptom, severe disease, and third COVID-19 wave (Table 4).

## Discussion

NLR value of 6.1 was found to be the threshold value for predicting the mortality of COVID-19 patients during admission. Higher NLR values were observed in patients who were 61 years old and above, males, those affected during the second wave of the outbreak, and those with hypertension. However, only age and the waves of the outbreak showed a significant association with NLR distribution. Also, higher values of NLR were found in a higher proportion among patients with severe and critical COVID-19, those who were admitted with mild and

**Table 3. Factors associated with severe disease in patients admitted with COVID-19 at KCMC Hospital: March 2020 to March 2022 (N = 504).**

| Variable | CRR (95% CI) | P-Value | ARR* (95% CI) | P-value |
|---|---|---|---|---|
| NLR | | | | |
| Normal NLR | 1 | | 1 | |
| Elevated NLR | 1.43 (1.29–1.58) | < 0.001 | 1.39 (1.26–1.55) | <0.001 |
| Age Groups (Years) | | | | |
| 18–30 | 1 | | 1 | |
| 31–60 | 1.11 (0.76–1.62) | 0.596 | 1.07 (0.75–1.53) | 0.713 |
| 61 and above | 1.23 (0.84–1.80) | 0.269 | 1.10 (0.78–1.56) | 0.59 |
| Sex | | | | |
| Male | 0.98 (0.89–1.10) | 0.841 | 1.02 (0.92–1.14) | 0.687 |
| Female | 1 | | 1 | |
| Comorbidity | | | | |
| Hypertension | 1.12 (1.00–1.25) | 0.044 | 1.12 (1–1.25) | 0.054 |
| Diabetes Mellitus | 1.04(0.93–1.17) | 0.453 | 1 (0.89–1.12) | 0.944 |
| HIV/AIDS | 0.68 (0.21–2.18) | 0.518 | 0.98 (0.87–1.1) | 0.747 |
| Chronic Kidney Disease | 0.88 (0.69–1.15) | 0.366 | 0.85 (0.66–1.1) | 0.228 |
| Other comorbidities | 0.69 (0.23–2.21) | 0.527 | 0.98 (0.89–1.12) | 0.821 |
| COVID-19 Waves | | | | |
| Wave 1 | 1 | | 1 | |
| Wave 2 | 1.32 (1.11–1.56) | 0.002 | 1.24 (1.04–1.47) | 0.015 |
| Wave 3 | 1.19 (0.98–1.44) | 0.081 | 1.13 (0.93–1.36) | 0.216 |
| Wave 4 | 1.30 (1.07–1.58) | 0.007 | 1.24 (1.02–1.50) | 0.029 |

Abbreviations: CRR, Crude Relative Risk. ARR, Adjusted Relative Risk. CI, Confidence Intervals. NLR, Neutrophil to Lymphocyte Ratio. HIV/AIDS, Human Immunodeficiency Virus / Acquired Immunodeficiency Syndrome. Adjusted risk ratios derived from the Modified Poison Regression Model (the generalized linear regression model with the Poisson family and log link function)

* Model adjusted for age groups, sex, co-morbidities, and COVID-19 waves.

moderate disease later deteriorated to severe form of illness, and those who died. Additionally, mortality risk increased with NLR increase. xx

In our study, we determined that an NLR cutoff value of 6.1 or higher is optimal for predicting mortality in COVID-19 patients, achieving a sensitivity of 92.1% and a specificity of 92.0%. These high values suggest that NLR is an important prognostic marker for identifying patients at an elevated risk of death. Upon comparing our findings with those from other regions, we noted variations in the optimal NLR cutoff values, which reflect differences in population demographics, disease severity, and healthcare settings across studies. For example, research from diverse geographic locations such as Israel and the USA reported NLR cutoff values ranging from 5.9 to 6.8, with corresponding sensitivities and specificities that varied considerably, illustrating the context-specific nature of NLR as a biomarker [27, 28]. However, an observational study conducted in Romania, identified a much higher optimal cutoff value of 9.1 for predicting mortality, with a sensitivity of 70% and a specificity of 67%, and an AUC of 68.9% [29]. The higher discrepancy observed in the Romanian study may be attributed to several factors. Firstly, its smaller sample size and the limited period covering only a single wave from May to October 2021 contrast with our study, which spanned four distinct COVID-19 waves. Additionally, demographic differences between Romania and Tanzania could have influenced the outcomes. These variations underscore the context-specific nature of NLR cutoff values in predicting mortality among COVID-19 patients. As Belgium study elaborated, such differences are crucial in understanding the observed disparities in NLR thresholds and their

**Table 4. NLR and other factors associated with in-hospital death AMONG patients admitted with COVID-19 at KCMC hospital (N = 504).**

| Variable | CHR (95%CI) | P-value | AHR (95%CI) | P-value |
|---|---|---|---|---|
| NLR | 1.03 (1.027–1.036) | <0.001 | 1.03 (1.021–1.038) | <0.001 |
| Age | | | | |
| 18–30 | 0.83 (0.385–1.777) | 0.63 | 1.68 (0.835–3.360) | 0.15 |
| 31–60 | 0.46 (0.331–0.648) | <0.001 | 0.53 (0.365–0.761) | <0.001 |
| >60 | 1.00 | | 1 | |
| Sex | | | | |
| Female | 1 | | 1 | |
| Male | 1.12 (0.838–1.501) | 0.44 | 1.08 (0.790–1.469) | 0.64 |
| Duration since first symptoms (days) | | | | |
| ≤7 | 1 | | 1 | |
| 8–14 | 0.51 (0.349–0.749) | 0.001 | 0.51 (0.341–0.758) | 0.001 |
| 15–21 | 0.39 (0.197–0.765) | 0.006 | 0.36 (0.188–0.699) | 0.002 |
| ≥22 | 0.21 (0.096–0.456) | <0.001 | 0.21 (0.093–0.339) | <0.001 |
| Disease severity | | | | |
| Mild and moderate | 1 | | 1 | |
| Severe and critical | 3.52 (2.162–5.715) | <0.001 | 2.74 (1.694–4.437) | <0.001 |
| COVID-19 Waves | | | | |
| Wave 1 | 1 | | 1 | |
| Wave 2 | 1.84 (1.175–2.880) | 0.01 | 1.10 (0.662–1.845) | 0.70 |
| Wave 3 | 2.29 (1.428–3.666) | 0.001 | 0.94 (0.547–1.611) | 0.82 |
| Wave 4 | 2.57 (1.553–4.252) | <0.001 | 1.87 (1.178–2.979) | 0.01 |
| Comorbidity | | | | |
| Diabetes (yes) | 1.14 (0.848–1.546) | 0.38 | | |
| Asthma/chronic lung disease (yes) | 1.03 (0.508–2.099) | 0.93 | | |
| HIV (yes) | 0.68 (0.251–1.828) | 0.44 | | |
| Other comorbidities (yes) | 0.69 (0.253–1.924) | 0.53 | | |
| Hypertension (yes) | 1.24 (0.930–1.660) | 0.14 | 1.10 (0.800–1.500) | 0.57 |

Abbreviations: CHR, Crude Hazard Ratio. AHR, Adjusted Hazard Ratio. CI, Confidence Intervals. NLR, Adjusted hazard ratios derived from the Modified Poison Regression Model (the generalized linear regression model with the Poisson family and log link function).

* Model adjusted for age groups, sex, co-morbidities, and COVID-19 waves.

predictive accuracy across different settings [28]. It is crucial to consider these factors when interpreting and applying NLR as a prognostic marker in different clinical settings.

Higher NLR values were significantly associated with an increase in disease severity. Specifically, patients with a baseline NLR of 6.1 and above were found to have a 1.43 times higher risk of developing severe and critical disease compared to those with a baseline NLR of less than 6.1. This association remained statistically significant even after adjusting for age, sex, COVID-19 wave, and specific comorbidities. As the severity of COVID-19 increased, the median NLR values were higher, indicating a positive correlation. These findings support the potential of NLR as an indicator of disease severity in COVID-19 patients. Several studies reported similar associations between NLR and disease severity in COVID-19 patients. For example, A study from China found higher NLR values in severe and critical cases compared to mild cases [30] while another study from Turkey demonstrated an increasing trend of NLR values with disease severity [31]. These studies collectively support the notion that NLR is associated with disease severity in COVID-19 [30, 31]. Meta-analyses also support the strong associations between elevated NLR and increased disease severity, further confirming the

significance of NLR in assessing disease severity [32, 33]. Despite variations in NLR values, the consistent trend of higher NLR values corresponding to increased disease severity indicates the potential utility of NLR as an indicator for assessing disease severity in COVID-19 patients.

A significant association between baseline NLR and the subsequent deterioration of COVID-19 patients with mild and moderate disease to severe and critical stages or death was observed. Among patients with mild and moderate disease who later deteriorated, those who were discharged alive had lower median NLR values compared to those who did not survive. Similar findings were reported in China where higher median NLR values at admission increased the risk of clinical deterioration [34].

Furthermore, on the association between NLR and mortality in patients with COVID-19, for each unit increase in NLR, the risk of in-hospital death increased by a factor of 1.03. This finding is supported by previous studies documenting higher in-hospital mortality risk for patients with higher NLR [35–39]

The underlying mechanisms explaining the observed outcomes can be attributed to the role of NLR as an indicator of systemic inflammation and immune dysregulation in COVID-19 [8–18]. Elevated NLR values reflect an imbalance between neutrophil and lymphocyte counts, suggesting an exaggerated inflammatory response and compromised immune function. This dysregulation can contribute to the severity of COVID-19 and increase the risk of adverse outcomes, including disease severity, clinical deterioration, and mortality [8–18].

## Conclusion

Our study demonstrated a significant association between the NLR and disease severity, clinical deterioration, and mortality in COVID-19 patients. Higher NLR values were positively associated with more severe disease and clinical deterioration. An NLR value of 6.1 or higher predicted higher mortality risk and had high sensitivity and specificity. These findings suggest the potential of NLR as a prognostic marker and highlight its potential as a reliable indicator for assessing disease severity, monitoring clinical progression, and identifying patients at higher risk of adverse outcomes, aiding in the effective management and treatment of COVID-19. Importantly, In this study, we excluded pregnant women, those missing baseline neutrophil and lymphocyte data, individuals with hematological malignancies, and those on chemotherapy, immunomodulating drugs, or long-term glucocorticoids, as these conditions can affect neutrophil and lymphocyte counts, leading to inaccurate NLR measurements. Other factors that can cause NLR derangement include infections, autoimmune disorders, chronic inflammation, physiological stress (e.g., trauma or surgery), and lifestyle factors like smoking, obesity, and alcohol use, as well as comorbidities such as cardiovascular disease, diabetes, and chronic kidney disease. While we focused on isolating COVID-19's impact on NLR, these factors should be considered when interpreting our findings, highlighting the complexity of NLR as a prognostic tool. Further research is warranted to explore the underlying mechanisms and validate the utility of NLR in larger and more diverse patient cohorts.

## Supporting information

**S1 Fig. The distribution of NLR among patients with mild and moderate disease in relation to deterioration to severe disease or death.**
(TIF)

**S2 Fig. The distribution of NLR across categories of disease severity in COVID-19 patients admitted at KCMC Hospital.**
(TIF)

**S3 Fig. Diagnostic accuracy of NLR on predicting mortality among confirmed COVID-19 Patients at KCMC Hospital: March 2020 to March 2022.**
(TIF)

## Acknowledgments

We acknowledge all doctors at KCMC who took part in data collection and all patients whose information enabled the availability of data used in this study. The authors also thank the KCMC isolation team for capturing these data in the electronic system.

## Author Contributions

**Conceptualization:** Norman Jonas Kyala, Elichilia Shao, Furaha Lyamuya.

**Data curation:** Norman Jonas Kyala.

**Formal analysis:** Norman Jonas Kyala.

**Investigation:** Norman Jonas Kyala.

**Methodology:** Norman Jonas Kyala, Innocent Mboya, Elichilia Shao, Laura Shirima, Furaha Lyamuya.

**Project administration:** Norman Jonas Kyala.

**Resources:** Norman Jonas Kyala.

**Software:** Norman Jonas Kyala.

**Supervision:** Norman Jonas Kyala, Innocent Mboya, Elichilia Shao, Francis Sakita, Kajiru Gadiel Kilonzo.

**Validation:** Norman Jonas Kyala, Kajiru Gadiel Kilonzo, Laura Shirima.

**Visualization:** Norman Jonas Kyala.

**Writing – original draft:** Norman Jonas Kyala, Innocent Mboya, Elichilia Shao, Kajiru Gadiel Kilonzo, Laura Shirima, Abid Sadiq, Damas Bayo, Daniel Mujuni, Furaha Lyamuya.

**Writing – review & editing:** Norman Jonas Kyala, Francis Sakita, Abid Sadiq, Elifuraha Mkwizu, Nyasatu Chamba, Annette Marandu, Sophia Muhali, Faryal Raza, Eliasa Ndale, Damas Bayo, Daniel Mujuni, Furaha Lyamuya.

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
