## [Decision Letter · Decision Letter 0]

20 May 2024

PONE-D-24-05173Neutrophil-to-Lymphocyte Ratio as a Prognostic Indicator in COVID-19: Evidence from a Northern Tanzanian CohortPLOS ONE

Dear Dr. Kyala,

Thank you for submitting your manuscript to PLOS ONE. After careful consideration, we feel that it has merit but does not fully meet PLOS ONE’s publication criteria as it currently stands. Therefore, we invite you to submit a revised version of the manuscript that addresses the points raised during the review process.

We look forward to receiving your revised manuscript.

Kind regards,

Jianhong Zhou

Staff Editor

PLOS ONE

Journal Requirements:

3. In the online submission form, you indicated that [Data cannot be shared publicly because of government guide. Data are available from the KCMC Institutional Data Access / Ethics Committee (contact via normanjonasmd@gmail.com) for researchers who meet the criteria for access to confidential data.]. 

Reviewers' comments:

Reviewer's Responses to Questions

**Comments to the Author**

1. Is the manuscript technically sound, and do the data support the conclusions?

Reviewer #1: Yes

Reviewer #2: Yes

2. Has the statistical analysis been performed appropriately and rigorously? 

Reviewer #1: I Don't Know

Reviewer #2: Yes

3. Have the authors made all data underlying the findings in their manuscript fully available?

Reviewer #1: Yes

Reviewer #2: Yes

4. Is the manuscript presented in an intelligible fashion and written in standard English?

Reviewer #1: Yes

Reviewer #2: Yes

5. Review Comments to the Author

Reviewer #1: Comments for Authors:

l would like to express our gratitude for the insightful study presented in your article.

methodology

1. l recommend providing further clarification on the utilization of ROC analysis for calculating sensitivity and specificity.

Discussion

1. Regarding the first paragraph, it would be beneficial to compare the mortality rate and severity of illness in COVID-19 patients between the Romanian study and the current study. Is the threshold for the Romanian study one and a half times that of the current study? Please interpret the reasons for such discrepancies and highlight any methodological differences between the two studies.

2. In the second paragraph, instead of repeating individual studies, summarizing and referencing similar studies on the positive correlation between Higher NLR and severe outcomes in COVID-19 patients would suffice.

3. Do you recommend healthcare professionals to assess NLR levels in COVID-19 positive patients during their visits?

4. Your study focused only on hospitalized individuals, while outpatient cases were not considered.

5. Is NLR testing useful a few days after symptom onset? Do you suggest periodic NLR testing?

6. The disparity in the population sizes between severe cases and milder cases in your study was notably low. Could this issue potentially impact the validity of the results?

7. As age increases, various underlying conditions may arise in individuals, which could be significant contributing factors to mortality in COVID-19.

Minor Comments:

• Consistency in referencing throughout the text according to the journal format is essential and requires correction for uniformity.

• The term "n=xxx" in Table 2 needs clarification for better understanding.

• Furthermore, the numbering and ordering of tables in the text are not consistent.

• The phrase "xx" in the first paragraph needs explanation for context.

Reviewer #2: I thank the journal editor for providing the opportunity to review this article. This article provided significant data on the significance of the NLR value as a prognostic indicator of the severity of COVID 19. However, there could be minor corrections to improvise the article. Point-wise suggestions are mentioned below:

1. Kindly include any reference in introduction suggesting impact of NLR with clinical or supportive care during the hospital stay (use of steroids or any supportive therapy's that effect the NLR ratio)

2. If there is any data on supportive care/therapy that impacted the NLR ratio in your results,?

3. Provide/brief the different time points at which the baseline and follow-up tests were performed (are any standard criteria followed?)

4. Look into line number 232 (mentioned as xx)

6. PLOS authors have the option to publish the peer review history of their article (what does this mean?). If published, this will include your full peer review and any attached files.

Reviewer #1: No

Reviewer #2: No

---

## [Author Response · Author response to Decision Letter 0]

12 Jul 2024

Dear reviewers and editors thank you for your insightful comments. All suggestions are appreciated and have been accommodated in the revised manuscript as it appears in attached file "response to reviewers".

---

## [Decision Letter · Decision Letter 1]

8 Aug 2024

PONE-D-24-05173R1Neutrophil-to-Lymphocyte Ratio as a Prognostic Indicator in COVID-19: Evidence from a Northern Tanzanian CohortPLOS ONE

Dear Dr. Kyala,

Thank you for submitting your manuscript to PLOS ONE. After careful consideration, we feel that it has merit but does not fully meet PLOS ONE’s publication criteria as it currently stands. Therefore, we invite you to submit a revised version of the manuscript that addresses the points raised during the review process.

We look forward to receiving your revised manuscript.

Kind regards,

Siddharth Gosavi, MBBS, MD Internal Medicine,DNB Internal Medicine

Academic Editor

PLOS ONE

Journal Requirements:

**Additional Editor Comments:**

Article is a good attempt. however other causes of NLR ratio derangement could have been elaborated better too. Please revise the article again.

Reviewers' comments:

Reviewer's Responses to Questions

**Comments to the Author**

1. If the authors have adequately addressed your comments raised in a previous round of review and you feel that this manuscript is now acceptable for publication, you may indicate that here to bypass the “Comments to the Author” section, enter your conflict of interest statement in the “Confidential to Editor” section, and submit your "Accept" recommendation.

Reviewer #1: All comments have been addressed

Reviewer #3: All comments have been addressed

Reviewer #4: (No Response)

2. Is the manuscript technically sound, and do the data support the conclusions?

Reviewer #1: Yes

Reviewer #3: Yes

Reviewer #4: Yes

3. Has the statistical analysis been performed appropriately and rigorously? 

Reviewer #1: Yes

Reviewer #3: Yes

Reviewer #4: Yes

4. Have the authors made all data underlying the findings in their manuscript fully available?

Reviewer #1: Yes

Reviewer #3: Yes

Reviewer #4: No

5. Is the manuscript presented in an intelligible fashion and written in standard English?

Reviewer #1: (No Response)

Reviewer #3: Yes

Reviewer #4: Yes

6. Review Comments to the Author

**Reviewer #1: **Thank you for the opportunity to review this manuscript. I appreciate the authors' efforts to address the comments and suggestions I provided in my previous review. The revisions have strengthened the manuscript significantly. Overall, I believe this manuscript has the potential to make a valuable contribution to the field.

Thank you for considering my review.

**Reviewer #3:** Although the topic is no longer "hot", the article could be interesting for some readers. The study can be important because of NLR's simplicity and cost-effectiveness compared to other inflammatory markers. Is it possible to specify how they evaluated the covid waves (which time intervals were included for delimitation) in the conditions where there were statistically significant values between the waves?Given that the NLR is increased in many other conditions, in the absence of comorbidity evaluation, not only hypertension, is there no risk of bias?it would be interesting to evaluate the severity of the disease depending on the waves of COVID. In the first wave, all the patients were probably hospitalized, possibly also the asymptomatic ones or those with mild symptoms

Even if the results of the study are not new and relevant from the point of view of the readers' interest, the paper is well written and the study is correctly carried out, and I think the opportunity for publication is up to the editorial team

R**eviewer #4: **This study has been conducted to find the association of NLR with COVID-19 mortality and deterioration. Main issues of the manuscript are:

1. Association of NLR with COVID-19 mortality and a worse outcome is already well published. Many articles as pointed out by the authors and many more are available looking at the same findings giving the same conclusion. It is not clear what the current manuscript adds to the already known body of literature. Authors have failed to highlight any novelty of the findings or any additional information in the current manuscript.

2. Although laboratory data has been extracted from patient records, there is no reference to the findings or any association of outcome/NLR with these parameters. Only the comorbidities have been analysed.

Current data does not have novelty and it is not clear what additional information is provided to the scientific community.

7. PLOS authors have the option to publish the peer review history of their article (what does this mean?). If published, this will include your full peer review and any attached files.

Reviewer #1: No

Reviewer #3: No

Reviewer #4: No

---

## [Author Response · Author response to Decision Letter 1]

26 Sep 2024

Dear editor;

I have uploaded the review as attachment.

---

## [Decision Letter · Decision Letter 2]

6 Nov 2024

Neutrophil-to-Lymphocyte Ratio as a Prognostic Indicator in COVID-19: Evidence from a Northern Tanzanian Cohort

PONE-D-24-05173R2

Dear Dr. Kyala,

We’re pleased to inform you that your manuscript has been judged scientifically suitable for publication and will be formally accepted for publication once it meets all outstanding technical requirements.

Kind regards,

Siddharth Gosavi, MBBS, MD Internal Medicine,DNB Internal Medicine

Academic Editor

PLOS ONE

Additional Editor Comments (optional):

I commend on you a good attempt at this article.

Reviewers' comments:

Reviewer's Responses to Questions

**Comments to the Author**

1. If the authors have adequately addressed your comments raised in a previous round of review and you feel that this manuscript is now acceptable for publication, you may indicate that here to bypass the “Comments to the Author” section, enter your conflict of interest statement in the “Confidential to Editor” section, and submit your "Accept" recommendation.

Reviewer #4: All comments have been addressed

Reviewer #5: (No Response)

2. Is the manuscript technically sound, and do the data support the conclusions?

Reviewer #4: Partly

Reviewer #5: Yes

3. Has the statistical analysis been performed appropriately and rigorously? 

Reviewer #4: Yes

Reviewer #5: No

4. Have the authors made all data underlying the findings in their manuscript fully available?

Reviewer #4: Yes

Reviewer #5: Yes

5. Is the manuscript presented in an intelligible fashion and written in standard English?

Reviewer #4: No

Reviewer #5: Yes

6. Review Comments to the Author

Reviewer #4: I thank the authors for addressing the comments during peer review. Please consider the following suggestions and make necessary changes.

1. Conclusions contain many limitations and information that does not give a conclusion to the paper or data presented in the manuscript. Limitations should be moved to the end of the discussion. Conclusion should be a concise summary of the findings derived from the paper and any recommendations. Please revise.

2. Attempt to improve the use of language.

3. Data availability statement needs revision

4. Author contributions is not complete. There are no authors given for the specified contribution.

5. Make sure that the references are correctly formatted according to the journal requirement.

Reviewer #5: I read with interest the manuscript submitted by Kyala and colleagues on the prognostic value of NLR in the context of COVID-19.

The manuscript further confirms the vast body of literature describing the clinical value of NLR in the management of COVID-19, yet from a resource-limited country. Such a scientific contribution is greatly appreciated, and further calls for policymakers to include NLR in the clinical guidelines on the management of COVID-19.

Few comments should be addressed before the manuscript is ready for publication:

- In several parts of the manuscript, the authors seem to cite a huge number of papers without proper justification or clear link. For example, the sentence in lines 96-98 describes NLR in various conditions in resource-limited settings like Tanzania, citing references 8-18. Some of these references are indeed on COVID-19 and almost none of them is from Tanzania.

- Moreover, lines 98-99 describes the cost effectiveness of NLR in COVID-19, which none of the referenced studies describe. Suggested and more relevant references would be as follows: (10.3390/healthcare10091780) (10.1136/bmjopen-2022-069493) (10.3390/diagnostics14171933).

- What is the rationale for using Kruskal-Wallis test as a comparative tool, instead of, for example a simple T tests? Please explain in the methods.

- Kindly mention the date of the issue of the IRB ethical approval.

- The discussion in its current form needs to be expanded by relating the current findings to the vast body of literature. For example, how does NLR compare to other diagnostic tools, e.g., SII? (10.3390/medicina60040602) (10.3390/vaccines12080861) and its use to develop new parameters (10.3390/molecules25235725) (10.3390/biomedicines11102649)/

- Please also include a separate paragraph at the end of the discussion to discuss the limitations of this study, and future research directions following the current study.

- The conclusion is way too long. Please consider shortening it.

7. PLOS authors have the option to publish the peer review history of their article (what does this mean?). If published, this will include your full peer review and any attached files.

Reviewer #4: **Yes: **Nilanka Perera

Reviewer #5: No

---

## [Editor Report · Acceptance letter]

13 Nov 2024

PONE-D-24-05173R2 

PLOS ONE

Dear Dr. Kyala, 

I'm pleased to inform you that your manuscript has been deemed suitable for publication in PLOS ONE. Congratulations! Your manuscript is now being handed over to our production team.

Kind regards, 

on behalf of

Dr. Siddharth Gosavi 

Academic Editor

PLOS ONE